# The feasibility of the virtually delivered dementia lifestyle intervention for getting healthy together (DELIGHT) program for people living with dementia and their family/friend care partners

Olivia L. Tupling[1], Bobby Neudorf [1]*, Sevana Haghverdian[1], Heather H. Keller[1,2], Carrie McAiney[2,3], Laura E. Middleton[1,2]

1 Kinesiology and Health Sciences, University of Waterloo, Ontario, Canada, 2 Schlegel-UW Research Institute for Aging, Waterloo, Ontario, Canada, 3 School of Public Health Sciences, University of Waterloo, Ontario, Canada

* bobby.neudorf@uwaterloo.ca

## Abstract

### Background

Improving supports to enhance quality of life for people living with dementia is a priority of research and practice. The DEmentia Lifestyle Intervention Program for Getting Healthy Together (DELIGHT) was co-designed by people with dementia, care partners, community stakeholders, and researchers with the goal of promoting 'living well' with dementia.

### Objective

A mixed methods study design was used to assess the feasibility of the 8-week virtual DELIGHT program, which incorporates group exercise and shared learning on topics related to health and wellbeing.

### Methods

Feasibility was evaluated through recruitment rate, attendance, and retention, along with semi-structured interviews that probed the program impact and recommended changes. Interviews were transcribed and analyzed using thematic analysis. Quality of life, physical activity, balance confidence, exercise self-efficacy, nutrition risk, social connection, balance, and strength were also assessed pre- and post-program.

### Results

A total of 19 participants were recruited to the two virtual program offerings (average of 4.75 participants/month). Of these, 17 (89.5%) completed the pre- and post-program evaluations. Average attendance was 78% demonstrating feasibility of the

**Data availability statement:** We have made our quantitative data available through figshare, which can be accessed through the following link: https://figshare.com/articles/dataset/DELIGHT_Quantitative_Data_-_RAW/29621477?file=56473256 However, the qualitative data contains sufficiently identifying information and the University of Waterloo Research Ethics Board has not given approval to share the interview transcripts. As a result, researchers who meet the criteria for access to confidential data will need to contact a the University of Waterloo Research Ethics Board to access the transcripts. Researchers will be able to contact the Waterloo Research Ethics Board by emailing researchoffice@uwaterloo.ca

**Funding:** Canadian Consortium on Neurodegeneration in Aging is supported by a grant from the Canadian Institutes of Health Research with funding from several partners, including the Alzheimer Society of Canada. The Canadian Institute of Health Research grant number is CNA-163902. The funders had no role in study design, data collection and analysis, decision to publish, or preparation of the manuscript.

**Competing interests:** The authors have declared that no competing interests exist.

program. Four themes related to the impact of and satisfaction with the DELIGHT program were identified: *Creating and Strengthening Connections* described the sense of community felt among participants and study leaders, *Sharing Knowledge and Learning* described participants enjoyment in learning from one another and sharing their own valuable knowledge, *Motivation to Improve Health and Wellbeing* described improvements in participants' functional abilities and motivation to continue healthy habits beyond the program, and *Providing Hope and Challenging Stigma* described a mindset shift to believing in the possibility of living well with dementia and challenging stigma.

## Conclusion

DELIGHT is a feasible lifestyle intervention for people living with dementia and their care partners. Further research to evaluate the feasibility of the in-person DELIGHT program and the effectiveness of both in-person and virtual offerings is warranted.

## Introduction

In 2019, there were more than fifty-five million people living with dementia worldwide [1]. At that time, total annual global health care costs associated with dementia were estimated to be USD $1.3 trillion [2]. Dementia is diagnosed when an individual experiences changes in their cognitive function and behaviour that are sufficient enough to impact their ability to function independently in daily life [3]. Dementia can also affect one's ability to participate in meaningful activities, disrupt social relationships, and lead to feelings of insecurity and loss of control [4]. Family members often assist with everyday activities, support medical care, and provide companionship and advocacy for the person they care for [5]. Both people living with dementia and care partners report reduced health-related quality of life and greater social isolation as compared to older adults without dementia or caring responsibilities [6–9].

Despite recent advances, there is no treatment to prevent or reverse dementia [10]. Identification of non-pharmacological strategies to support people living with dementia and their care partners to live well post-diagnosis are critical priorities [11,12]. Living well with a chronic illness or disability is defined by the Institute of Medicine as "the best achievable state of health that encompasses all dimensions of physical, mental, and social wellbeing" [13]. The IDEAL study framed living well with dementia as a composite measure of quality of life, wellbeing, and life satisfaction. Among people living with dementia, everyday functioning, physical health and fitness, social networks and supportive relationships, and mental and emotional wellbeing are all predictors of living well [14].

Non-pharmacological interventions may improve many of the factors associated with living well with dementia [14]. For example, exercise has been shown to improve physical and cognitive function, including executive function and attention [15,16]. Exercise can also improve mental wellbeing and self-esteem, and reduce depressive symptoms [17,18]. Good nutrition is also vital to maintaining health, independence,

and the wellbeing of older adults, including people living with dementia [19]. Challenges with eating, drinking, and maintaining weight become more common as dementia progresses, which may contribute to malnutrition and subsequent functional loss [20].

Group programs that support healthy living may have additional benefits over adoption of these lifestyle interventions on one's own. Social connection and a sense of belonging are essential to wellbeing [21]. Purposeful connections and active involvement in meaningful activities are important for people living with dementia and their care partners [14]. Connection with others experiencing dementia may help individuals to feel less alone and facilitate sharing of information and strategies to promote health and function and overcome challenges associated with dementia [22].

Identifying effective interventions to support the wellbeing of people living with dementia and their care partners has been identified as a research priority [11]. The DEmentia Lifestyle Intervention for Getting Healthy Together (DELIGHT) program was co-designed by people living with dementia, care partners, community service providers, health care professionals, and researchers in the spring of 2020 [23]. DELIGHT is a multicomponent lifestyle program to promote living well with dementia, targeted to people living with dementia and their family/friend care partners. DELIGHT is a group program that includes exercise and shared learning that addresses topics related to healthy living (healthy eating, physical activity, social engagement, mental wellbeing, sleep quality). When the DELIGHT program was co-designed, the team was primarily focused on an in-person program, recognizing that development of a virtual option for greater reach was a long-term goal. However, the ongoing public health restrictions put into place to combat the spread of COVID-19 [24,25], required suspension of in-person programs [26]. As a result, we accelerated the development of the virtual DELIGHT program and piloted the virtual program before the in-person program. Hence, the purpose of this feasibility pilot study was to: (1) determine the feasibility of the virtual DELIGHT program among people living with dementia and family care partners; and (2) understand the impact, strengths, challenges of the program and suggested changes for future delivery from the perspectives of program participants, staff, and volunteers.

## Materials and methods

### Study design and setting

This mixed methods study assessed the feasibility of the 8-week, virtual DELIGHT program among people living with dementia and their family/friend care partners across two offerings in the fall of 2021 (October 5th – November 25th) and spring of 2022 (April 25th – June 15th). In this phase, the feasibility of the program and outcomes were assessed in a smaller scale to help inform future large-scale trials. The DELIGHT program was delivered from the University of Waterloo, but participants were dispersed geographically across Canada. Feasibility was assessed using feasibility indicators (recruitment rate, attendance, retention), staff and volunteer reflections, and an end-of-program interview. Outcomes of the future effectiveness evaluation were also assessed before and after the intervention. This study was approved by the University of Waterloo Research Ethics Committee (REB #43178). All participants provided verbal informed consent.

### Participants

People living with dementia were eligible for the study if: (1) they had sufficient English proficiency to engage in and understand the study information letter, shared learning, and exercise instructions; (2) identified themselves as a person living with dementia or mild cognitive impairment or a care partner to a person with one of these conditions; (3) had access to the internet and a computer with a camera; and (4) were safe to exercise, as determined by completion of the Get Active Questionnaire and, if indicated, physician clearance. Exclusion criteria included: (1) insufficient English to communicate with study leaders and other program participants; (2) contraindications to exercise; and (3) severe cognitive impairment (i.e., unable to follow two-step commands, or provide informed consent).

Participants were recruited through social and traditional media, email lists and newsletters, advertisements in the Alzheimer's Society program guides, institutional websites, and word of mouth. Individuals interested in the study contacted study staff via phone or email. They were given the opportunity to ask questions. If still interested, they were screened for eligibility.

## Safety considerations

All study personnel were trained on emergency procedures prior to the program (see Supplementary Material S1 for detailed emergency procedures). During an introductory Zoom session prior to the program start, participants provided their personal and emergency contact information and health information (e.g., medical conditions and medications). Each participant's contact number was tested to ensure it worked. The study leader and the participants discussed the space where the participant planned to exercise and the study leader made recommendations for changes if needed (e.g., choosing a location with non-slippery floors, removal of rugs/clutter that posed a tripping hazard).

Personal and emergency contact information was on-hand during all sessions. At the beginning of each session, the program leader reviewed the safety checklist—that is, to wear comfortable clothes and running shoes and have water bottle, sturdy chair, and phone within arm's reach. They were reminded to work at their own pace and to immediately let the study leader know if they felt unwell (e.g., if they were dizzy or felt chest pain) or if they had any changes to their health. Participants were asked to leave their cameras on throughout the exercise session. Exercises were monitored by the volunteers and Registered Kinesiologist, and participants received feedback to ensure safe movement. Participants were to notify the study leader if they were going to leave the screen by waving or giving a thumbs up. If participants left the screen without notice, a research volunteer phoned them.

## Intervention

The DELIGHT program scope, outcomes, components, and structure were developed by the co-design team. "Living well" was designated as the primary goal of the program [27]. Quality of life, physical and mental health, function, mood, and social engagement and support are all components of living well. DELIGHT is an 8-week program with two 1.5-hour sessions per week. The program was intentionally designed to be fun, social and flexible within a core format and guiding principles [27]. Each session should include 50–60 minutes of moderate aerobic, resistance and balance exercises, and 20–30 minutes of shared learning on nutrition and other health topics (with an option to share a healthy snack) [27].

The virtual DELIGHT program was delivered on Zoom. We accommodated a minimum class size of 6 participants and a maximum class size of 12 participants. A Zoom link was sent to participants the morning of each session, to ensure participants accessed the correct link each time. Each virtual session started with a review of the safety checklist (e.g., wearing proper footwear, have a phone within reach), followed by approximately 50 minutes of group exercise and 30 minutes of facilitated, shared learning with a 10-minute break between. An orientation to session-specific use of Zoom was provided in the first session so that participants knew how to mute/unmute, turn the camera on and off, and adjust the camera angle.

Participants were mailed a study and reflection booklet as well as a resistance band. The study booklet included the program schedule, DELIGHT learning resources (resource sheets, healthy recipes), instructions for using resistance bands, an exercise guide with images and descriptions, and links to additional dementia friendly resources (for example, Ontario Brain Institute's Physical Activity and Alzheimer's Disease Toolkit [28]). The reflection booklet provided space for participants to reflect on what they learned, program elements that they enjoyed or found challenging, and recommended changes for future offerings.

The exercise portion of the program included a 5- to 10-minute warm-up, approximately 30 minutes of conditioning (aerobic exercise, strength training, and balance exercises) and a 5- to 10-minute cool-down. The virtual exercise was structured as an 8-exercise circuit that was repeated twice each session with 3–5 minutes of aerobic exercise between

each round. Strength training exercises were designed to target large muscle groups and included four lower body exercises and one chest, back, shoulder and bicep exercise. Strength training used body weight or resistance bands to increase load. A Registered Kinesiologist developed and supervised all exercise sessions and met with the research team weekly to ensure a high quality and consistent approach. All exercises were demonstrated at two levels, one seated and one standing. (See Supplementary Material S2 for a sample). Further modifications were offered as needed. Both exercise leaders were pinned to the screen on Zoom. Participants were asked to stay within a moderate intensity range (rating of perceived exertion of 3–6 on the Borg 10-point scale [29]). Participants reported their rating of perceived exertion using their fingers every four exercises. Everyone but the primary study leader was muted to ensure audio clarity and minimize distractions. Participants could unmute at any time if they wished to speak. A research assistant volunteer monitored the class and took notes (e.g., about adherence to the exercise program, challenges or adverse events).

The shared learning sessions were connected to a DELIGHT program resource used to guide semi-structured discussion. Resource topics focused on nutrition, physical activity, sleep, mental and emotional wellbeing, and social engagement and support. Each resource provided high level information about the topic, spaces to record reflections, and referral to other resources for additional information. Shared learning was facilitated by the study leader, driven by the participants, and supported by volunteers. Participants were encouraged to record and reflect on the resource and their experience participating in the shared learning in their reflection booklets. A research assistant volunteer monitored the discussion and took notes (e.g., moments of silence, questions that arose, ideas or concerns that stood out).

## Assessments

Assessments were conducted within two-weeks of the start and end of the intervention. The movement assessment was conducted by the program leader as observations and results from the movement assessment informed the exercise program and options recommended for the participant. All other assessments were conducted by an assessor not involved in intervention delivery.

## Descriptive characteristics

Participants reported gender (man, women, non-binary, other), age, place of residence and type of dwelling, ethno-racial identity, highest level of education, and time of diagnosis for those with dementia (this year, less than five years ago, more than five years ago).

**Primary outcome: feasibility.** Quantitative feasibility outcomes included: (a) recruitment rate; (b) attendance; and (c) retention with targets set before the study start. Not meeting these targets would indicate a need to reflect on and make adaptations to the study and/or program processes prior to further evaluation. **Recruitment** numbers and time needed for recruitment was noted by a research assistant. Our feasibility target for recruitment was at least six participants recruited per month (i.e., that we could recruit the minimum required for a program offering within the month). **Attendance** was tracked by a research assistant (off-screen). Our feasibility target for attendance was an average of 75% of sessions attended across participants. Our feasibility target for **retention** was that at least 80% of participants completed the program and post-program assessments.

Feasibility was also assessed through open-ended reflections by participants, volunteers, and study leaders. Each group was encouraged to make notes in the reflection booklet after each session. Separate spaces were provided to note what went well, what did not go well, and recommended changes. The reflection booklets were sent back to the research team after the program ended.

An assessor conducted interviews with each participant, program leader, and volunteer within two weeks following the last DELIGHT session. Interviews used a semi-structured interview guide, with 5 or 6 questions about their experience

in the program and addressed challenges and supports needed, the impact of the program, and recommendations to improve future offerings of DELIGHT. Interviews took approximately 30-minutes. People living with dementia were interviewed separately from their care partner unless otherwise requested.

**Secondary outcomes: preliminary effectiveness.** Effectiveness assessments were conducted within a two-week period before and after the program. These may be potential outcome measures for a future, larger scale study. Post-program, the interviews and effectiveness assessments were usually conducted in the same session.

**Quality of life.** The Dementia Quality of Life scale (DEMQOL) was used to measure quality of life for all participants [30,31], including people living with dementia and care partners. Although the DEMQOL is not typically used with care partners, most of the probes are sufficiently generic to apply to care partners. The specific probes around memory would still be relevant to people aware of cognitive issues and their impact and not only people living with dementia. DEMQOL is a 28-item, interview-administered, dementia-specific tool to assess health related quality of life [30,31]. DEMQOL questions ask participants to rate their feelings, memory and thinking abilities, everyday life, and overall quality of life over the last week using a 4-point Likert scale with possible scores ranging from 28–112; higher scores indicate better quality of life [30,31].

**Movement assessment.** Lower body strength was assessed using a timed, five-time sit-to-stand. Balance was assessed using a stance test. The ability of participants to hold each of three stances (feet together, semi-tandem stance, and tandem stance) was timed up to a maximum of ten seconds. The participants' performance and observations during these tasks were used to select the exercise starting point (sitting or standing) for each participant.

**Physical activity.** The Physical Activity Scale for the Elderly (PASE) questionnaire was used to assess physical activity levels [31]. PASE asks participants to report on their leisure, physical, household and work associated activities within the last seven days [32]. PASE has been recommended as one of the best tools for assessing physical activity levels amongst older adults in a recent review [33]. It has been used in previous research among people living with dementia [34].

**Balance confidence.** The Activities-specified Balance Confidence (ABC) scale was used to assess participants' balance confidence and is valid in persons with MCI [35,36]. Participants rate their confidence in not losing their balance in a variety of daily activities on a scale of 0–100% [35]. The total score is calculated by summing item scores and dividing by number of items, total scores can range from 0–100% [35].

**Nutrition risk.** Participants were screened for nutrition risk using the 14-item Seniors in the Community: Risk Evaluation for Eating and Nutrition (SCREEN-14) questionnaire [37]. SCREEN-14 asks participants to report on their current eating habits, including challenges with chewing and swallowing, and any changes in weight. Each question has multiple rated response options, a total score can be summed and ranges from 0–64, with a score of less than 50 representing high nutrition risk [37]. SCREEN-14 was validated against dietitians' nutritional assessment [37].

**Exercise self-efficacy.** We created a self-efficacy for physical activity scale relevant to this study, adapting from the SCI Exercise Self-Efficacy Scale (ESES) [38]. We used a series of six statements that focused on participants' knowledge and confidence related to physical activity and exercise. The first four statements were drawn from the SCI-ESES, with minor wording changes (i.e., *I am confident that I can find enjoyable ways to be physically active most days of the week, set and accomplish realistic physical activity/exercise goals, be physically active/exercise with no access to a gym/ rehabilitation facility, be physically active without the direction from an exercise therapist/trainer* [38]. Additional phrases were added to better capture the Canadian experiences with weather changes (*I am confident that I can find enjoyable ways to be physically active in poor weather conditions (such as rain or snow))* and participants' confidence in their knowledge about physical activity and exercise *(I understand the benefits of physical activity/exercise)*. Participants rated their confidence on a 4-point Likert scale, with the total score reflecting the sum of responses.

**Social Connections and Loneliness.** Two measures of social isolation and loneliness were added to the second virtual offering of DELIGHT based on interview findings from the first cohort. The Friendship scale was used to measure social isolation, conceptualized as the absence of social support [39]. The Friendship scale is a 6-item questionnaire

asking participants to report how often they experience a feeling on a five-point scale from *not at all* to *almost always*. Lower scores indicate higher levels of social isolation.

The UCLA three-item Loneliness scale, was used to measure participant's feelings of loneliness [40]. The three questions asked participants how often they felt *left out, a lack of companionship, and isolated from others*. Participants rate each question on a three-point scale from *hardly ever* (1) to *often* (3). Responses can be summed to generate a final score; higher scores indicate higher feelings of loneliness; lower scores represent a more desirable outcome.

## Analysis

Demographics, feasibility indicators, and outcome assessments were described using mean and standard deviation, median and range, or % [n], as appropriate. The feasibility indicators were compared to the pre-determined targets. Effectiveness outcomes were described at baseline and post-program but were not analyzed as this feasibility study was not powered to detect change in outcomes.

Interviews were audio-recorded, cleaned, and transcribed verbatim by a research assistant. Reflection booklet responses were added to the end of the interview transcript of each participant, program leader, or volunteer. Qualitative data were analyzed using thematic analysis following the steps recommended by Braun and Clarke [41]. An inductive approach was used to identify themes and parallel content analysis was used to deductively identify issues related to feasibility (challenges and recommendations).

A team of three researchers independently read and coded the interview and reflection transcripts using NVivo V.14. Each transcript was reviewed and coded by at least two researchers as some members of this group were new to qualitative analysis. Researchers took notes on common phrases and patterns to develop initial themes. The three researchers came together to discuss and develop themes and create rich descriptions for each. Themes were discussed and reviewed by the research team, including experts in nutrition, physical activity, and public health, and final themes were verified by team discussion and review of draft findings. The findings include data extracts (quotes) that supported each theme.

**Assessing Trustworthiness.** Researchers' experiences, beliefs and biases influence the methodology used and interpretation of the results [42]. In this study, the lead researcher is white woman working under a constructivist paradigm. Constructivists believe that each individual constructs their own concept of reality through their cognition and past experiences [43]. The secondary coders were a white man and a South-Asian woman.

Trustworthiness was supported through efforts to promote credibility, transferability, dependability, and confirmability [44]. To promote credibility, investigator triangulation was used, whereby multiple researchers independently read and coded each transcript before sharing and discussing ideas. The analysis team also had repeated debriefing sessions with co-investigators and advisors to encourage opportunities for peer scrutiny [44]. Data triangulation resulted from analyzing both interview and reflection responses, which helped connect immediate experiences during the program with interview responses [44]. Furthermore, a researcher who was not involved in the delivery of the program facilitated all interviews to help encourage participants to speak freely about both positive and negative aspects of their experience in DELIGHT. Rich descriptions of the study setting, contextual information, methods, and analysis were provided to enhance contexts to which findings may transfer [44]. To promote dependability, an audit trail was established by keeping detailed descriptions of reflexive thoughts throughout all phases of the analysis [44]. Detailed descriptions of study findings, paired with relevant data extracts (participant quotes) to support themes were provided to enhance confirmability.

## Results

### Participants

Participant characteristics are shown in Table 1. There were nineteen participants (12 people living with dementia, 7 care partners) in the virtual pilots of the DELIGHT program, seven participated in the fall of 2021 and 12 participated in the

**Table 1. Participant characteristics by participant type, (mean [range] or n [%]).**

| | Person living with dementia (n = 12) | Care partners (n = 7) |
|---|---|---|
| **Age (years), mean (range)** | 74 (57-89) | 68 (36-88) |
| **Gender, n (%)** | | |
| Women | 7 (58%) | 6 (86%) |
| **Education, n (%)** | | |
| College/University degree or higher | 6 (50%) | 5 (71%) |
| Technical, vocational or apprenticeship | 4 (33%) | 2 (29%) |
| Highschool Diploma | 2 (17%) | |
| **Ethnicity, n (%)** | | |
| White | 10 (83%) | 6 (86%) |
| South Asian | 2 (17%) | 1 (14%) |
| **Province, n (%)** | | |
| Ontario | 7 (58%) | 4 (57%) |
| British Columbia | 2 (17%) | 1 (14%) |
| New Brunswick | 2 (17%) | 1 (14%) |
| Alberta | 1 (8%) | 1 (14%) |
| **Residence, n (%)** | | |
| Private home | 8 (67%) | 6 (86%) |
| Apartment/condo | 3 (25%) | 1 (14%) |
| Seniors Residence | 1 (8%) | 0 (0%) |
| Lived alone | 2 (17%) | 1 (14%)[1] |
| Lived with partner | 7 (58%) | 5 (71%) |
| Lived with partner and child | 2 (17%) | 1 (14%) |
| Lived with child | 1 (8%) | |
| Diagnosis, n (%) | | |
| More than 5 years ago | 1 (8%)[1] | |
| Less than 5 years ago | 4 (33%) | |
| This year (2021) | 6 (50%) | |
| No official diagnosis | 1 (8%) | |

[1]Percentages don't add up to 100% due to rounding.

spring of 2022. Participants joined from four provinces across Canada from the east to the west coast. The mean age of people living with dementia and care partners was 74 years (57–89 years) and 68 years (36–81 years), respectively. There were more women than men participating and more than 80% of participants identified as white.

**Primary outcome: feasibility**

**Recruitment:** The overall recruitment rate was 19 people over 4-months of recruitment, falling short of our recruitment target. Seven participants were recruited over a two-month period in the first offering, which was lower than the pre-determined feasibility target (6 people in one month). In the second offering, additional recruitment strategies were added (notably, a print media article) and, subsequently, twelve participants were recruited over a two-month period, reaching our feasibility target. It is difficult to determine how many people living with dementia may have received advertisement of the program, however, 62 individuals responded through mass email or other forms of advertisement. Those who did not

participate, either did not respond or were lost after follow-up (n = 23), were ineligible (n = 15), or were only interested in an in-person program (n = 5). Of note, there were more people interested in the second offering than could be accommodated (n = 14 on the waiting list).

**Attendance:** The average attendance over both offerings was 78%, meeting the pre-determined target for feasibility of 75%. Reasons for missed sessions included having a prior commitment (n = 5; for example, a medical appointment or a wedding), feeling unwell (n = 5), and experiencing technical issues (n = 4).

**Retention:** Seventeen of nineteen participants (89.5%) completed the program and assessments, meeting the pre-determined target for feasibility of at least 80% retention. Two participants dropped out of the study, one due to a diagnosis of chronic obstructive pulmonary disease and the second due to challenging interpersonal dynamics between the person living with dementia and their care partner.

### Secondary outcomes: preliminary effectiveness

Participant effectiveness outcomes are described in Table 2. Any changes from before to after the intervention should be considered preliminary; statistical comparisons were not conducted. Positive trends were observed for several outcomes. Most notably, the change for the mean DEMQOL score was 6-points, meeting the criteria for a minimally important difference (≥5 points) [45].

### Qualitative findings

All participants and study personnel described enjoying their participation in the virtual DELIGHT program and that they would be interested in participating again. Four themes related to the impact of and satisfaction with the DELIGHT program were identified: *Creating and Strengthening Connections, Sharing Knowledge and Learning, Motivation to Improve Health and Wellbeing,* and *Providing Hope and Challenging Stigma.*

**Table 2. Outcome assessments at baseline and post-program, presented as group median (min-max) (n = 17).**

| Assessment Tool | Pre-DELIGHT Group Median (Range) | Post-DELIGHT Group Median (Range) |
|---|---|---|
| **All Completers** | **(n = 17)** | **(n = 17)** |
| Five-Time Sit-to-Stand Test (sec.) | 14.3 (8.5-24.5) | 13.1 (8.4-19.7) |
| Tandem Stance (sec.) | 10.0 (5.1-10.0) | 10.0 (2.0-10.0) |
| Dementia Quality of Life (DEMQOL) | 90.0 (63.0-105.0) | 96.0 (55.0-111.0) |
| Physical Activity Scale for the Elderly (PASE) | 96.0 (8.6-268.5) | 115.6 (8.3-216.5) |
| Activities-specified Balance Confidence (ABC) scale (%) | 93.3 (65.0-100.0) | 91.9 (34.3-99.4) |
| Seniors in the Community: Risk Evaluation for Eating and Nutrition (SCREEN-14) Questionnaire | 38.0 (31.0-52.0) | 36.5 (27.0-57.0) |
| Modified Exercise Self-Efficacy Scale (ESES) | 20.0 (10.0-24.0) | 20.5 (9.0-23.0) |
| **Spring Completers Only** | **(n = 11)** | **(n = 11)** |
| The Friendship Scale | 23.0 (15.0-30.0) | 28.0 (20.0-30.0) |
| The Loneliness Scale | 4.0 (3.0-8.0) | 3.5 (3.0-9.0) |

**Theme 1: Creating and strengthening connections.** Participants enjoyed the sense of community created with others who shared similar experiences with dementia and described DELIGHT as a positive social outlet. DELIGHT fostered a supportive environment that promoted comfort and confidence among participants. Participants described feeling "less alone," a "sense of comfort," and a "sense of reassurance" because of the connections they created within the program and their observations of how other participants were affected by, and coped with, dementia. One person living with dementia said:

> I was listening to somebody and it's like I was less alone because I was around people, so I felt less, I felt connected, [...] that we were all, we were all in the same boat. (PLWD001)

Participants' comfort within the group evolved over the 8-week program, with many describing initial feelings of nervousness and anxiety related to group exercise and shared learning with "strangers." They noted that they had stepped out of their comfort zone and felt joy from being present with others, discussing relevant challenges and ideas, and exercising together. One care partner said:

> I was really surprised at his [husband living with dementia] reaction of going outside his comfort zone and, yeah, which is very odd for him to go outside his comfort zone but there was a trust in the safety thing there, that must have developed somehow over the weeks. (CP017)

Not only were new connections made, but in some cases, pre-existing relationships appear to have been strengthened from the DELIGHT program. Another care partner mentioned the benefit of having a regularly scheduled appointment to spend time with their spouse during the day, as they would normally not have that time together, and described it as "time well spent."

**Theme 2: Sharing knowledge and learning.** Participants learned from each other during shared learning, writing down others' coping strategies, favourite healthy food options, supportive resources, and more. Participants appreciated the knowledge and ideas that others put forward. Participants had extensive and diverse experience with cooking, physical activity, and other healthy behaviours and felt they had valuable knowledge to contribute, which promoted a sense of independence and confidence. One person living with dementia said:

> I've got a lot of experience in cooking and all that kind of jazz behind me – but there was some new things, and listening to other people discuss, within our half-hour of discussion, I did learn things. (PLWD029)

The program booklet and fact sheets were described as useful and informative resources. Participants asked thought-provoking questions to get further detail on points of interest and used reflection space to take notes. Others were keen to contribute to conversation and answer questions, using real life experiences and providing practical strategies. For example, strategies shared included healthy, pre-packaged meal options and how getting outdoors for sunlight had improved one person's sleep. One care partner said:

> I found it very informative. There were a number of things through the discussion that came up that I wasn't aware of that I really appreciated. Especially the audience, I guess you'd call participants would put forward in the, in their discussions. And I did write those things down. (CP008)

Participants suggested that the design of the program booklet facilitated this learning through use of reflection and providing space on fact sheets to take notes.

**Theme 3: Motivating to improve health and wellbeing.** Participants noted changes in their abilities and habits that reflected improvements in their health and wellbeing. Participants described an increased awareness of their current

eating, physical activity, sleep, and social habits and reported using the practical strategies introduced in the shared learning to incorporate healthy changes to their daily routine. Participants also noticed physical improvements in fitness and strength that translated to better ability to perform activities of daily living and participation in hobbies.

Reinforced and new knowledge gained through shared learning increased participants' awareness of their daily habits and encouraged them to make healthier choices. Participants started questioning their eating habits and many reflected on the "ordinary" foods in their diet, thinking about how they could make small adjustments to increase nutrition. One care partner said:

*And about nutrition, I really appreciated that because sometimes I thought oh, I'm eating healthy and then I'm questioning, maybe it's not as healthy as I thought it was. And how can I make the adjustments. And it was nice to see that there were, there was input as to how you could modify and add to your regular routine. (CP008)*

Another participant talked about incorporating tuna into her and her husband's diet, using the program to ask for tips and develop a better understanding of the nutritional benefits of specific foods. Other participants mentioned "knowing better" but that they had been getting away with whatever would fill their stomach. DELIGHT reminded them of the importance of good nutrition, not only for their bodies but also for their minds. One person living with dementia said:

*I want to stay in my apartment as long as I can. So, I realised how – DELIGHT program includes diet and exercise, how much more important that is, even though I knew it, to keep that possible. If I'm not fit, I can't take care of myself. If I don't eat I will be sick eventually. And even though I knew that, the DELIGHT program reminded me of it. (PLWD010)*

DELIGHT also increased participants' awareness of their physical activity habits and, for many, inspired motivation to continue with physical activity outside of the DELIGHT sessions. Participants mentioned reflecting on sessions and noticing they felt "good," "energized" and "happier" afterwards. Participants mentioned sleeping better on days the DELIGHT sessions were held and feeling hopeful to have more control over their sleep by incorporating more physical activity and time outdoors into their daily routine. A care partner noted that she typically spent the morning sitting and knitting and often fell asleep part way through. She now tries to spend time outside gardening in the morning and noticed improvements in her energy levels throughout the day. Participants spoke about the motivation to incorporate more physical activity into their regular routine, describing the program as a "kick in the pants" to get up and move.

If participants were unable to attend one of the DELIGHT sessions, many said they would do the exercise portion on their own time using the bands and the exercise guide that was provided in the program booklet. One care partner stated:

*After we'd missed something, we did it just on our own. And sometimes, when we're watching the news or something on telly, because we get ads […] I would do some of the exercises that we'd learned from [DELIGHT], during those breaks. (CP023)*

Physical improvements in strength and fitness translated to increased ability to perform activities of daily living and continue with active hobbies. Some participants noticed changes in weight, with one mentioning he had lost five pounds over the course of the program -- a positive for him. One person living with dementia said:

*I went for an X-ray today and I was in a huge area, normally I walk with a walker, and I didn't today – I'm sure that I have gained some more strength by doing this as well, and that's really important to me. (PLWD029)*

A participant talked about how he used to love shooting archery bows and had lost the strength to pull the bows back far enough to shoot the arrow. By the end of the program, he was able to shoot arrows again, his care partner said:

*When he [husband with dementia] came back he said, "guess what, I strung and I could pull back two of my bows" he couldn't do that anymore a few weeks ago. And I said, "[husband] it's because you're doing exercises with the bands, and it's strengthened your muscles." And it was so nice to see him so excited about being able to do that. (CP044)*

**Theme 4: Providing hope and challenging stigma.** DELIGHT inspired hope amongst participants by challenging social, structural, and self-stigma of dementia. Participants reported that they were seeing life with dementia differently and felt motivated to live in the moment and make the most of each day. Participants felt that they were improving their future by being physically active and learning from the program. They were also excited to be involved in the DELIGHT pilot, reflecting that they felt they were a part of something that will be beneficial for other people living with dementia.

Two participants disclosed their traumatic experiences when diagnosed with dementia, highlighting the negative effects of stigma in the healthcare system. Participants described that there were few activities and services where they felt "safe" or "normal". They described DELIGHT as the first program where they felt accepted, where program and study leaders recognized them as unique individuals without any assumptions or stereotypes. They highlighted the key role that study leaders and volunteers have in carrying forward the intentions of the program and how their positive attitudes and perceptions were felt and adopted by participants. DELIGHT was specifically described as a program that would be beneficial to have access to immediately after a diagnosis, challenging the stigma within the healthcare system and how the system is run. One person living with dementia said:

*The first year I was told I had two to six years to live, and that there was nothing – the doctor just sent me on my way. And the first year I didn't know anything, so I was just waiting to die. So, if somebody, if the doctors, every doctor in Canada would have that, that this would be available and they could say, "Here, start doing this." Because when they're diagnosed it would be a world of difference. (PLWD001)*

After seeing the other participants actively engaged in DELIGHT and observing their own improvements, participants started to challenge their own self-stigma. Participants described previous behaviours of self-isolation, not eating, and not being physically active because they didn't think it was as important any more or they thought "what's the use." One person living with dementia noted:

*I think I kind of gave up and although I know the importance of exercise, I didn't think it really applied to me that much anymore. Now, every day I exercise in one way shape or form. (PLWD010)*

Participants also described making a choice to live in the moment and were now looking forward to every day, even the basic tasks of cooking, cleaning, and moving about. One person living with dementia talked about how DELIGHT had changed her attitude, and how she had started feeling more motivated to live:

*I have a new zest for life. Even if I walk down the road and see people, it's great. It's given me, inspired me to live a little better. Yeah, not think about the end. Think about what's here now. (PLWD010)*

Participants also described a shift in their sense of agency, feeling they had more control over their wellbeing by incorporating healthy eating, physical activity, and other practical strategies into their routine. Many participants talked about "knowing better" however, they found they would make excuses to not do things. DELIGHT made them more aware of these behaviours and made them feel they had the power to make positive changes.

Care partners also described a shift from thinking they had to be the sole caregiver to understanding that there are other resources that they can and should access. One care partner's view shifted from thinking that she always had to be

the one to fix things, as there were few dementia-friendly supports available in their rural neighbourhood, to understanding that there could be other places for support. She stated:

*I think it [DELIGHT] changed more my view on society's acceptance of it [dementia], and awareness of that. And uhm, maybe I don't have to always be the person that tries to fix things. […] I think I felt, you know, there's lots of things that can be different for us in our environment with our family I, it was nice to see that it could be different outside our environment, and it be okay. (CP017)*

Other care partners also spoke about this mindset shift towards recognizing that there were outside resources and supports, and that reaching out for help and taking the time to care for themselves is not selfish. One care partner recognized that she needs to care for herself to be the best version of herself to care for her husband.

DELIGHT was also seen as a steppingstone for participants to become engaged with further resources within their communities, as many disclosed, they were not aware of or would not have reached out to dementia-friendly resources without taking part in DELIGHT. DELIGHT was viewed as an introduction, with the hope that participants could incorporate healthy habits into their lives and continue building upon them once the program was over. Participants appreciated the additional resources that were available through the program booklet and imbedded video links, and often inquired about further resources via email after the session was over. One care partner said:

*…being involved in the program, and just knowing that there are things all the time being done, gives me hope that things will evolve, and there'll be lots of things to access, and be able to benefit from as [spouse with dementia's] journey sort of progresses. (CP027)*

### Challenges & adaptations

Through content analysis of the participant, study leader, and volunteer interviews, challenges were identified along with strategies and recommendations to improve the virtual DELIGHT program going forward. Challenges and recommendations are described in Table 3.

## Discussion

The purpose of this study was to examine the feasibility of the virtual DELIGHT program for people living with dementia and their family/friend care partners, and to understand the perceived impact, strengths and challenges, and suggestions for improving the virtual DELIGHT program. Our results suggest that the virtual DELIGHT program was an enjoyable and feasible lifestyle intervention for people living with dementia and their care partners. Participants formed connections and were able to share knowledge and learn from one another. They recognized improvements in their health and wellbeing and were motivated to continue with healthy habits beyond the program. These experiences with the program encouraged hope among participants, helping overcome stigma and inspiring them to be mindful and live each day to the fullest.

The overarching aim of the DELIGHT program is to improve the likelihood and extent of living well with dementia among both people living with dementia and care partners. Themes from the reflections of participants, staff, and volunteers suggested that the program created feelings of hope, empowerment, and connection, and that these helped participants overcome prior feelings of stigma from others and their own stigma of dementia. The shared learning in the DELIGHT program gave participants a sense that they (and others like them) had valuable knowledge to contribute. The role of program leaders and volunteers may have played a part in the positive experiences of the participants as they helped to create a warm and inviting environment. The program may not have the same effect with a leader and volunteers who held negative or stigmatized perceptions of dementia. Perceived negative attitudes toward dementia among

**Table 3. Perceived Challenges and Recommended Adaptations to Overcome.**

| Challenge | Strategies to Overcome |
|---|---|
| **Ensuring program is appropriate for participants with diverse abilities.** | |
| • Participants who were more active at baseline were less likely to perceive benefits from the virtual exercise program, finding the exercises too easy.<br>• Participants who had difficulty following instructions were more likely to require support from their care partner, leading to potential care partner stress. | • Have further options and modifications for exercise intensity.<br>• Group participants in programs based on their physical and cognitive abilities.<br>• Have smaller class sizes. |
| **Balancing voices during shared learning** | |
| • Some participants were keen to contribute and others more hesitant.<br>• Participants reported feeling frustrated when they had difficulty jumping into the conversation.<br>• Study leaders found it difficult to allow all participants to share. | • Leaders gently cut off dominant voices.<br>• Leaders use longer pauses to wait for responses.<br>• Leaders invite contributions from quieter members.<br>• Use Zoom features such as raise your hand or breakout rooms |
| **Providing individualised feedback during exercise** | |
| • It was difficult for study leaders to demonstrate and give instructions, while also monitoring participants and giving individual corrections, especially in the 12-person class. | • Provide written instructions or videos for exercises beforehand so participants know what to expect and can ask questions.<br>• Consider having smaller class limit. |
| **Challenges with technology** | |
| • It was difficult for leaders to see the full body of participants as exercise position varied.<br>• Participants reported that it was difficult to see the study leader if the device was far enough away so the leader could see their full body.<br>• Trouble shooting technology while leading exercise or shared learning was difficult. | • Having a designated technology support person to trouble shoot with participants in a separate breakout room.<br>• Have troubleshooting instructions available for all devices (iPad, laptop, tablet, TV)<br>• Send an email reminder to participants prior to each session with the Zoom link. |
| **Instructor recommendations to further support social connection** | |
| • Have time for participants to connect informally outside of DELIGHT sessions (including before or after the session).<br>• Facilitate exchange of contact information for willing participants. | |

health and social care professionals may serve as a barrier for people living with dementia and their care partners to access services [46].

Participants described feeling a sense of community within the DELIGHT program and feeling less alone in the post-program interviews. These interview findings prompted us to include the Friendship scale [39], and ULCA three-item Loneliness scale in the second offering [40]. Both showed average group improvement. Connecting socially with others living with dementia has been reported to increase feelings of self-worth and usefulness, supporting empowerment among people living with dementia [47], which aligns with our results. Identifying with others, or feeling group identity, may help increase confidence, and amplifies the voices of those taking part, allowing participants to learn from each other and adopt a shared narrative [48]. Furthermore, social connectedness and participation in meaningful social activity are domains that have been described as important to quality of life among people living with dementia [31,49,50]. Social support may further be associated with improved self-esteem and quality of life [51]. In agreement, preliminary description of quality of life before and after the DELIGHT program suggests that improvements in quality of life, as assessed with the DEMQOL, may be clinically meaningful [45].

Participants felt empowered to improve their health through health behaviours, within the program and beyond. Engaging in DELIGHT gave participants the experience, knowledge, and tools to make healthier choices and feel more in control of their health. Having a choice and control have been identified as components of empowerment among people living with dementia [52]. In this study, participants reported feeling more positive and energized after DELIGHT sessions, giving them motivation them to continue with physical activity and healthy eating beyond the program, this aligned with the group's average improvement on the Physical Activity Scale for the Elderly [32]. Participants also noted positive impact

to their functional strength and fitness, observed by improvements in activities of daily living and meaningful hobbies. Improvements in function and quality of life have been observed following both exercise interventions and nutrition interventions among people with dementia [53,54], aligning with our results.

The structure of the DELIGHT program also appears to be feasible and acceptable. Almost 90% of participants completed the 8-week program, with attendance surpassing the pre-determined target of 75%. These results are comparable to previous studies of virtual exercise programs for people living with dementia and their care partners [53,55,56]. Attendance for the virtual delivery of the DELIGHT program was slightly less compared to similar community-based in-person programs [57,58]; however, many absences were due to health appointments, prior commitments, or illness, which were not considered study related. There were a few instances of problems with technology, some of which influenced attendance (n = 4) and have been noted as a challenge of virtual programs [53,55]. Without a care partner present, virtual programs may not be as accessible for those with more severe cognitive impairment or those who have challenges with accessing and using the zoom platform.

Recruitment was the greatest challenge to feasibility, with rates meeting feasibility targets only in the second offering. Recruitment in previous published virtual exercise interventions for people living with cognitive impairment or dementia has primarily relied on existing patient or research populations [53,55,56,59,60], whereas DELIGHT recruited participants new to the research group and study. Both DELIGHT pilot offerings occurred during periods of COVID-related public health restrictions when many traditional recruitment strategies, including targeted promotion in clinics, senior centres and community organizations that serve people living with dementia, were not possible or effective due to greatly reduced or closures of services [61]. Our most successful recruitment strategy was an article in a local newspaper. After the article publication, many people expressed interest in the DELIGHT Program within a week of the article being published (n = 26), more than could be accommodated in the spring 2022 offering. This demonstrates that there was considerable interest in such a program when potential participants can be reached.

Although our study included participants with diverse ages (36–89 years) and geographic locations (Atlantic to Pacific coast), our study population was predominantly white. Data from the 2021 census showed that approximately one in seven Canadian seniors are a visible minority [62]. The ability to recruit and deliver an exercise program virtually may reduce health inequities due to geography [63,64]. With a push to widen internet access, virtual programs can add supports for people in rural populations [61]. Additional efforts to build trust and relationships with underserved ethno-racial groups may be needed to improve reach [61]. Further research will need to examine whether the program resonates with and is feasible for different ethno-racial groups.

Accommodating participants' diverse abilities, skills, and personalities was seen as challenging in the virtual DELIGHT program. People who are more fit may need more challenging exercise options, and some participants may need more encouragement or opportunity to share in shared learning. Previous virtual programs for people living with dementia limited their class sizes to 6–8 participants to minimize interference during shared learning and optimize viewability [55,57]. Zoom breakout rooms have also been used to support individualized exercise modification and progression [57]. Those with more severe cognitive impairment may need additional supports from their care partners or others. Indeed, inclusion of care partners has been recommended in previous virtual exercise studies among people living with dementia to support safety [55,65,66]. Smaller class sizes may be warranted for future delivery as program leaders could then provide more individualized exercises and supports. In addition, having a dedicated person to provide technology support may relieve some of the duties from the program staff. Indeed, a dedicated support person was recommended in another virtual exercise and activity program for people living with dementia and care partners [66].

Participating in virtual programs from home reduces common barriers to physical activity such as transportation, distance, and time [65,66], which is particularly important for those living in rural locations [63]. Virtual programs may also be a more accessible option in the Canadian winter weather. In an in-person setting it may be easier to provide participants with exercises and feedback tailored to their individual needs, potentially providing further benefit to physical

function and independence. In-person programming may also give space for further one-to-one conversation and connection and attract participants within a local setting, potentially leading to lasting connections beyond the program. The structure of the DELIGHT program is inherently flexible so it can be delivered in diverse settings with different equipment and resources available. Future research will explore how to implement the DELIGHT program and train leaders across Canada. In practice, the next phase will be a multi-centre, single arm study dictated in part by funding streams available that target this type of program.

The results of this study add to a small, yet growing literature focused on investigating virtually delivered exercise interventions for people living with dementia and their care partners. Some potential limitations must be acknowledged. Participants were not required to provide documentation of a dementia diagnosis or complete a cognitive assessment to participate in the program. Receiving a dementia diagnosis can be challenging in Canada due to wait times, location or culturally appropriate screening, whereas cognitive assessments can feel intrusive and stressful, countering the purpose of a program to promote health and well-being. Furthermore, participants were required to have access to a computer or tablet with a webcam and internet connection to participate, so feasibility of the virtual DELIGHT program may be limited to individuals with higher levels of technological proficiency. A quantitative measure of program satisfaction and fidelity were not used, and may be efficient and useful in subsequent, larger scale phases of the study. Information bias is possible, as the majority of the assessments were self-reported. Furthermore, when people living with dementia and care partners completed the post-program interview together, they may have biased their responses positively for the benefit of their partner. However, assessments in couples only occurred in a minority of participant dyads.

## Conclusion

The virtual DEmentia Lifestyle Intervention for Getting Healthy Together (DELIGHT) program is a feasible lifestyle intervention for people living with dementia and their care partners who are familiar with virtual meeting platforms and have appropriate technology. Ongoing recruitment efforts are needed to support program enrolment. Further research will evaluate the feasibility of the in-person DELIGHT program and the effectiveness of both approaches in diverse settings. Based on these findings further questionnaires on sense of belonging and hope would be useful additions to demonstrate outcomes. It is likely that some adaptations, or strengthened relationships, are needed to attract more diverse ethno-cultural groups to the program.

## Supporting information

**S1 File. Emergency procedures.**
(DOCX)

**S2 File. Example 8-exercise circuit.**
(DOCX)

## Author contributions

**Conceptualization:** Heather H. Keller, Carrie McAiney, Laura E. Middleton.

**Formal analysis:** Olivia L. Tupling, Bobby Neudorf.

**Funding acquisition:** Heather H. Keller, Carrie McAiney, Laura E. Middleton.

**Investigation:** Heather H. Keller, Carrie McAiney, Laura E. Middleton.

**Methodology:** Olivia L. Tupling, Sevana Haghverdian.

**Project administration:** Sevana Haghverdian.

**Supervision:** Laura E. Middleton.

**Writing – original draft:** Olivia L. Tupling.

**Writing – review & editing:** Bobby Neudorf, Sevana Haghverdian, Heather H. Keller, Carrie McAiney, Laura E. Middleton.

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
