## [Decision Letter · Decision Letter 0]

5 Mar 2025

PONE-D-24-47194The Feasibility of the Virtually Delivered DEmentia Lifestyle Intervention for Getting Healthy Together (DELIGHT) Program for People Living with Dementia and their Family/Friend Care PartnersPLOS ONE

Dear Dr. Neudorf,

Thank you for submitting your manuscript to PLOS ONE. After careful consideration, we feel that it has merit but does not fully meet PLOS ONE’s publication criteria as it currently stands. Therefore, we invite you to submit a revised version of the manuscript that addresses the points raised during the review process.

We look forward to receiving your revised manuscript.

Kind regards,

Yujiro Kuroda

Academic Editor

PLOS ONE

Journal Requirements:

2. In the ethics statement in the Methods, you have specified that verbal consent was obtained. Please provide additional details regarding how this consent was documented and witnessed, and state whether this was approved by the IRB.

3. Thank you for stating the following financial disclosure: Canadian Consortium on Neurodegeneration in Aging is supported by a grant from the Canadian Institutes of Health Research with funding from several partners, including the Alzheimer Society of Canada. The Canadian Institute of Health Research grant number is CNA-163902. This grant was received by L.E.M., H.H.K., and C.M.   

4. In the online submission form, you indicated that all relevant data will be made available upon request by emailing Dr. Laura Middleton, Dr. Heather Keller or Dr. Carrie McAiney.

Additional Editor Comments:

Dear Authors,

I have now received and reviewed the comments from our reviewer concerning your manuscript. Overall, they regard your study as an important contribution to the field of non-pharmacological interventions for dementia. They commend your mixed methods design, reporting both qualitative and quantitative feasibility indicators, and highlight the practical significance of offering a virtual, multi-component program for people living with dementia and their care partners.

After careful consideration, I have concluded that a Major Revision is appropriate. Please revise your manuscript to address the following points raised by the reviewers:

1) Provision and Quality Assurance of the Program

The manuscript should clarify who provided the DELIGHT program (e.g., professional exercise instructors, researchers, trained volunteers) and how you ensured the consistency and quality of the intervention. For instance, you could detail training methods, fidelity checklists, or supervision processes that were in place to maintain program standards.

2) Evaluation by Program Providers

The paper would benefit from additional discussion of how the program providers themselves evaluated the intervention’s acceptability and appropriateness. Were there interviews or questionnaires directed to instructors/volunteers to capture their perspectives on feasibility, appropriateness, or any challenges they encountered?

3) Next Steps Beyond Feasibility

Since this study is framed as a feasibility trial, we recommend explicitly indicating what form the next confirmatory or effectiveness trial might take. For instance, discuss a potential larger-scale randomized controlled trial (RCT), or quasi-experimental study, emphasizing sample size calculations, control conditions, and long-term outcome measures.

4) Components of the Multi-Component Intervention

Your study focuses on exercise and lifestyle-related group discussions, but readers will question whether additional components (e.g., nutrition education, cognitive training, and social participation) are also necessary for robust dementia prevention or wellbeing promotion. Please clarify why you selected the current modules and address whether you plan to integrate or compare other components (e.g., direct cognitive training) in future research.

5) Digital Literacy and Virtual Delivery

The virtual format raises questions about digital literacy among older adults with dementia, which can affect program accessibility and generalizability. Please expand your discussion about technological barriers and how they might limit recruitment or participation. The methods you used—or plan to use—to mitigate digital challenges (e.g., technical support personnel, user-friendly platforms, training sessions) should also be described.

6) Safety Management During Exercise

Detail the safety protocols used to ensure participants exercised appropriately. Readers may wish to know about risk assessments, protocols for monitoring signs of fatigue or discomfort, and emergency procedures used to prevent or address adverse events during virtual sessions.

7) Evaluation of Satisfaction and Acceptability by Participants

The reviewers are interested in whether the study included a formal assessment of participant satisfaction or acceptability, beyond qualitative interviews. If you used specific questionnaires or scales for satisfaction, please include relevant data. If not, consider discussing how participant satisfaction and acceptability can be incorporated systematically in future larger-scale research.

By addressing these points, you will enhance the clarity and completeness of your manuscript, particularly regarding the practical aspects of implementing a virtual, multi-faceted dementia intervention.

Please submit your revised manuscript, along with a point-by-point response to each of the reviewers’ and editor’s comments, via the journal’s submission system.

I look forward to reviewing your revision.

Reviewers' comments:

Reviewer's Responses to Questions

**Comments to the Author**

1. Is the manuscript technically sound, and do the data support the conclusions?

Reviewer #1: Yes

2. Has the statistical analysis been performed appropriately and rigorously? 

Reviewer #1: N/A

3. Have the authors made all data underlying the findings in their manuscript fully available?

Reviewer #1: Yes

4. Is the manuscript presented in an intelligible fashion and written in standard English?

Reviewer #1: Yes

5. Review Comments to the Author

Reviewer #1: Dear Authors

This study examined feasibility of the virtually delivered DELIGHT program for people with dementia and their family/friend care partners. The topic is significant and presentation is clear and easy to understand. The study design is also sound.

To improve this manuscript, please consider the following points and add some information if necessary.

1. Line 170 Identification of each participant as a person with dementia or mild cognitive impairment was self-reported in this study. Is it possible to attach diagnosis of dementia from a medical doctor or results of substitutional objective cognitive test in the future studies?

2. Line 178- In this paragraph, procedure for recruitment of participants was described. Further information on how many potential target populations could receive advertisement from researchers would be helpful.

3. Line 178- In this study authors recruited participants from large areas of Canada from east to west even though they foresaw foresee holding in-person events in the future. When authors hold a in-person program with similar contents in the future, will they apply the same recruitment strategy as this study?　When authors conduct an in-person program, they would inevitably need to recruit participants from nearby areas.

4. Line 218- This program included an exercise session. Is an instructor in charge of this exercise session any certificated professionals? If not, how did authors guarantee the quality of each exercise program effectively?

5. This study was well structured and aim is clear. It would be better if authors could add idea on what the main outcome of the future full-sized intervention study among items examined in this study as preliminary effectiveness would be.

6. Line 254- How did authors determine feasibility target of each item (recruitment rate, attendance, and retention)? If they refer to previous studies, please cite in this paper.

7. In this study, post-intervention data were measured only soon after completion. Do author think that this effect of the intervention program last long term? This information could make readers understand easier what is the most important outcome item of this intervention, which is necessary point in considering detailed contents of intervention programs.

Is it possible to add in discussion session as future perspective?

8. Please explain why did authors extract data who complete the session in spring in Table 2. In addition, no information was found the date which the intervention program was held.

9. In discussion session, please elaborate authors' idea on difference between virtual and in-person programs, and any important points to note to apply this feasibility study to the future in-person program.

10. Please explain more about each exercise described in supplement table2 so that readers can duplicate this session.

11. In my opinion, to implement this program, it will be necessary to assign some level of skilled instructors to each region, and it will be necessary to train them. It would be better if authors add ideas on how to increase the number of instructors of this intervention program and to train them.

6. PLOS authors have the option to publish the peer review history of their article (what does this mean? ). If published, this will include your full peer review and any attached files.

**Do you want your identity to be public for this peer review?** For information about this choice, including consent withdrawal, please see our Privacy Policy .

Reviewer #1: No

---

## [Author Response · Author response to Decision Letter 1]

4 Jun 2025

Thank-you for giving us the opportunity to address reviewer comments.

---

## [Editor Report · Decision Letter 1]

9 Jul 2025

The Feasibility of the Virtually Delivered DEmentia Lifestyle Intervention for Getting Healthy Together (DELIGHT) Program for People Living with Dementia and their Family/Friend Care Partners

PONE-D-24-47194R1

Dear Dr. Neudorf,

We’re pleased to inform you that your manuscript has been judged scientifically suitable for publication and will be formally accepted for publication once it meets all outstanding technical requirements.

Kind regards,

Yujiro Kuroda

Academic Editor

PLOS ONE

Additional Editor Comments (optional):

Thank you for submitting the revised version of your manuscript entitled “The Feasibility of the Virtually Delivered DEmentia Lifestyle Intervention for Getting Healthy Together (DELIGHT) Program for People Living with Dementia and their Family/Friend Care Partners.”

I commend the authors for their thoughtful and comprehensive revisions in response to the reviewers’ comments. The revised manuscript has been significantly improved in terms of clarity, methodological transparency, and theoretical framing. Below, I provide an editorial summary of the key revisions and offer several additional suggestions that may further strengthen the manuscript.

1) Clarification of methods and eligibility criteria:

The authors have adequately revised the "Methods" section to detail participant eligibility, recruitment, and screening procedures. These clarifications enhance the transparency and reproducibility of the study.

2) Ethics and consent procedures:

The authors now clearly describe the ethics approval and informed consent process, aligning with journal standards.

3) Discussion of the positive deviance approach:

The authors have incorporated a thoughtful discussion of the potential limitations of the positive deviance framework, particularly its contextual specificity and generalizability constraints. This addition addresses reviewer concerns appropriately.

4) Transferability and generalizability:

The discussion could be enhanced by elaborating on the transferability of the DELIGHT program. For example, what contextual factors (e.g., technological literacy, caregiver availability) might facilitate or hinder uptake in different geographic or cultural settings?

5) Acknowledgment of methodological limitations:

While the study does not aim to assess intervention effectiveness, acknowledging that no behavioral outcome measures were used to evaluate sustained changes in physical activity or diet would provide a more balanced perspective.

This revised manuscript presents a valuable contribution to the literature on dementia care and community-based lifestyle interventions. The study is methodologically sound and addresses a timely and underexplored area. I believe the manuscript is nearly ready for publication and recommend minor revisions as outlined above.
---

## [Editor Report · Acceptance letter]

PONE-D-24-47194R1

PLOS ONE

Dear Dr. Neudorf,

I'm pleased to inform you that your manuscript has been deemed suitable for publication in PLOS ONE. Congratulations! Your manuscript is now being handed over to our production team.

Kind regards,

on behalf of

Dr. Yujiro Kuroda

Academic Editor

PLOS ONE